# Hand Hygiene, Face Mask Use, and Associated Factors during the COVID-19 Pandemic among the Students of Mongar Higher Secondary School, Bhutan: A Cross-Sectional Study

**DOI:** 10.3390/ijerph20021058

**Published:** 2023-01-06

**Authors:** Tashi Wangchuk, Ugyen Wangdi, Ugyen Tshering, Kinley Wangdi

**Affiliations:** 1Mongar Higher Secondary School, Mongar 43002, Bhutan; 2Faculty of Education, University of Canberra, Bruce, Canberra, ACT 2617, Australia; 3National Centre for Epidemiology and Population Health, College of Health and Medicine, Australian National University, Acton, Canberra, ACT 2601, Australia

**Keywords:** Bhutan, hand hygiene, face mask, students, use, modelling, factors, Mongar Higher Secondary School

## Abstract

Non-pharmacological measures, such as hand hygiene and face mask use, continue to play an important role in the fight against the COVID-19 pandemic. However, there is a paucity of studies on the adherence to these measures among students in Bhutan. Therefore, we aimed to investigate hand hygiene and face mask-wearing behaviours, as well as their associated factors, among the students of Mongar Higher Secondary School, Bhutan. We conducted a cross-sectional study amongst the students of Mongar Higher Secondary School in Bhutan. The students self-answered the questionnaire on web-based Google Forms. Multivariable logistic regression for good hand washing and face mask use was conducted in order to identify statistically significant socio-demographic covariates. The correlation between hand hygiene and mask use was investigated using Pearson’s correlation coefficient. A total of 533 students completed the survey questionnaire, 52.9% (282) of whom were female students. Facebook (44.3%, 236) and TV (35.5%, 189) were the two most popular sources of information on COVID-19 prevention and control. Good (scores of ≥80% of total scores) hand hygiene and face mask use were reported in 33.6% (179) and 22.1% (118) of students. In multivariable logistic regression, male students presented 79% (adjusted odds ratio [AOR] = 1.79, 95% confidence interval [CI] = 1.23–2.613) odds of engaging in good hand hygiene, compared to female students. Compared to grade 9, those in grade 10 were 60% (AOR = 0.4, 95% CI 0.158–0.998) less likely to engage in good hand hygiene. Boarding students presented 68% (AOR = 1.68, 95% CI 1.001, 2.813) higher odds of wearing a face mask compared to day students. There was a significant positive correlation between good hand hygiene and face mask use (r = 0.3671, *p*-value < 0.001). Good hand hygiene and face mask use were reported in less than one-third of the study participants. It is recommended to continue educating students on good hand hygiene and face mask use through popular information sources.

## 1. Introduction

The COVID-19 pandemic, a disease caused by the SARS-CoV2 virus which originated in Wuhan, China, has wreaked havoc, disrupting the livelihood of billions of people across the world. The total number of cases has crossed 651.9 million, with 6.6 million deaths as of 21 December 2022 [1]. In addition to the millions of lives lost due to COVID-19, the pandemic continues to have a tremendous impact on both the mental and physical health of various individuals [2]. Despite the rolling out of vaccines, newer strains (e.g., Omicron sub-lineages) are emerging, resulting in new waves of COVID-19 infections [3,4,5,6,7,8,9,10,11]. In addition to vaccination, non-pharmacological measures, such as washing hands, face mask wearing, and social distancing are expected to continue to play an important role in controlling the COVID-19 pandemic [12,13,14]. Face mask use has a greater role in reducing COVID-19 due to aerosol transmission of it. However, hand hygiene can have added health benefits by reducing other infections.

Bhutan has reported 62,524 COVID-19 cases with 21 deaths, and 2 million vaccine doses administered to date [15]. In addition, the Bhutan government has continued to advocate non-pharmacological prevention measures, including hand washing, social distancing, and the use of face masks in public places [16,17]. Facilities for hand washing have been set up in schools and institutes across the country [18,19]; however, the success of preventive measures initiated by the government depends on the uptake and adherence to these preventive measures [20,21]. No studies have been undertaken in order to understand hand hygiene and face mask use in Bhutan, including among high school students. In other parts of the world, there have been varying reports of adherence to hand hygiene and face mask use in the general population [22,23] and students [24,25,26]. It is important to review and evaluate the adherence to the education program in different population groups.

As COVID-19 is a highly contagious disease primarily affecting the respiratory system, places such as schools still pose a significant risk of SARS-CoV-2 transmission [27]. However, transmission can be averted through adequate preventive measures [28,29] and adherence to these measures. Therefore, in this study, we aim to understand the current situation of hand hygiene and face mask use among Mongar Higher Secondary School students in Eastern Bhutan. The study findings will be useful to inform policymakers and healthcare professionals, regarding the development of future public health interventions, awareness raising, and health education programs.

## 2. Methods

### 2.1. Study Design and Setting

We conducted a cross-sectional study among the students of Mongar Higher Secondary School in classes 9 to 12, enrolled for the 2022 academic year. Mongar Higher Secondary School was selected through convenience sampling [30]. The survey was carried out between June and July 2022.

### 2.2. Sample Size

The required sample size for this study was calculated as follows:(1)$$n=\frac{Z^{2}P\left(1-P\right)}{d^{2}\;},$$
where *n* is the sample size; *Z* is the value of the statistic in a normal distribution for a 95% confidence interval (this value is 1.96); *P* is the expected proportion (with a total proportion of one), set at 0.5; *d* is the precision (with a total proportion of one) set at 0.05.

The sample size was 384, allowing for a 15% dropout rate; the final sample size was 441. However, all students in classes 9 to 12 were invited to the study. There were a total of 558 students for the 2022 academic year: 79 in class 9, 51 in class 10, 274 in class 11, and 154 in class 12.

### 2.3. Inclusion Criteria

Inclusion criteria were: (i) Students in classes 9–12; (ii) of either gender and (iii) students enrolled for the 2022 academic year.

### 2.4. Exclusion Criteria

Exclusion criteria were: (i) Students under class 9 and (ii) students not volunteering to fill out the questionnaire.

### 2.5. Data Collection Instruments

We used a web-based self-administered questionnaire, in order to minimise the transmission of COVID-19. The survey questionnaire was developed in the Google survey tool (Google Forms). The link generated from the Google Form was circulated to the students through their class teachers. The link led to the first page of the Google Form, which summarised the research background, aims, and expected outcomes. At the end of the first page, a declaration of confidentiality and informed consent to voluntarily participate in the study was provided. Only upon agreeing to participate in the study was the main questionnaire opened. 

The questionnaire was adapted from an earlier study [31] and WHO guidelines [32] and consisted of four parts. Part I included sociodemographic questions (consisting of seven questions), Part II focused on knowledge of COVID-19 and relevant sources of information (four questions), Part III on hand hygiene (with 11 questions), and Part IV included questions on face mask use (16 questions). The survey was managed by the three teachers (T.W., U.W., and K.) from the school, who are part of the study team. It took around 20–30 min for students to answer the questionnaire.

### 2.6. Data Analysis

The responses for hand washing and face mask wearing were assessed using a 4-point Likert scale. The scoring for correct answers was 3, 2, 1, and 0 for always, often, rarely, and never, respectively. In the case of negatively quoted questions, reverse scoring was used: 0, 1, 2, and 3 for always, often, rarely, and never, respectively. Maximum scores of 33 and 45 could be scored for hand washing (Part III) and face mask wearing (Part IV), respectively. A final score of ≥26.4 and ≥38.4 was considered to classify as good hand washing and good face mask wearing (1 = good, 0 = poor), respectively. This classification was based on the modified Bloom’s cut-off point [33].

Data were extracted into MS Excel 2016 (Microsoft Corporation), double-checked, and validated for accuracy. Descriptive analysis was conducted using frequencies and proportions for categorical variables. Univariate and multivariable logistic regression models for good hand hygiene and face mask use were utilised to identify statistically significant socio-demographic covariates. Any variable with a *p*-value < 0.2 in the univariate analysis was considered a candidate variable in the multivariable logistic model. Adjusted odds ratios (AOR) with 95% confidence intervals (CI) were used to determine the correlates of each independent variable with all potential dependent variables in the full model, with a *p*-value ≤ 0.05 considered statistically significant. The correlation between hand hygiene and mask use was investigated using Pearson’s correlation coefficient. All explanatory variables in the multivariable model were tested for multicollinearity using a variance inflation factor (VIF), where VIF < 10 was considered a good fit for regression analysis (see Appendix A). Statistical analysis was conducted using the STATA version 16 software (Stata Corporation, College Station, TX, USA).

## 3. Results

A total of 533 students completed the survey questionnaire with a response rate of 95.5%. There were 52.9% (282) female students, 61.6% (326) stayed in the school hostel (boarding students), and 53.3% (283) and 28.3% (150) were students in grades 11 and 12, respectively. More than half 56.3% (299) of the student’s fathers had no formal education. Similarly, 76.1% (405) of their mothers had received no formal education. Farmers were the most common occupation of both fathers (55.9%, 292) and mothers (52.0%, 275) of the study participants (Table 1). The most common sources of information on COVID-19 prevention and control were: Facebook (44.3%, 236), TV (35.5%, 189), other social media (23.1%, 123), and teachers (20.5%, 109); see Figure 1. The two most commonly used face masks were medical surgical masks (64.4%, 342) and non-medical fabric masks (22.2%, 118), respectively (Figure 2).

The mean score of hand hygiene was 24.8 (range 6–33). However, only 33.6% (179) reported good hand hygiene. A higher percentage of male students reported good hand hygiene, compared to female students (55.9% vs. 44.1%; Table 1). The mean score of face mask use was 31.3 (range 15–45). Good face mask use was reported by 22.1% (118) of the study participants. Within each grade, grades 9 (45.3%, 24) and 12 (35.3%, 53) reported good hand hygiene, compared to grades 10 (24.4%, 11) and 11 (31.8%, 90), while good face mask use was highest in grade 11 (25.1%, 71); see Appendix A.

In the multivariable logistic regression, male students presented 79% (AOR = 1.79, 95% CI 1.230–2.613) odds of engaging in good hand hygiene, compared to female students. Grades 10, 11, and 12 were less likely to engage in good hand hygiene, compared to grade 9; however, the result was significant only for grade 10 (AOR = 0.4, 95% CI 0.158–0.998; Table 2). In terms of good face mask use, boarding students (i.e., students staying in the school hostel) presented 68% (AOR = 1.68, 95% CI 1.001–2.813) higher odds, when compared to day students (i.e., students residing in the school hostel); see Table 3. There was a significant positive correlation between good hand use and mask use (r = 0.3671, *p*-value < 0.001; Table 4).

## 4. Discussion

This is the first study in Bhutan to evaluate hand hygiene and mask use among students in the context of the COVID-19 pandemic. Good (scores of ≥80% in total scores) hand hygiene and face mask use was reported in only 33.6% and 22.1% of study participants, respectively. Good hand hygiene was associated with being male, while those in grade 10 were less likely to engage in good hand hygiene. Boarding students were more likely to use face masks consistently, compared to day scholar students. There was a positive correlation between good hand hygiene and face mask use.

Only one-third of participants reported good hand hygiene. This finding is contrary to what has been observed during the initial phase of the pandemic, where people showed good practices regarding COVID-19 [34]. However, similar findings of poor adherence to good hand hygiene have been reported in students in China [31] and the general population in other parts of the world [35,36]. Since the start of the pandemic, the Royal Government of Bhutan has educated the public on good hand hygiene through all available mass media, including newspapers, television, radio, and various social media platforms (e.g., Facebook). At the start of the pandemic, people started panic-buying various items, such as face masks, sanitisers, and other essential food and grocery items [37]. As a result, hand sanitisers were distributed free-of-charge by the government. Contrary to what was observed during the initial phase of the COVID-19 pandemic, the lower percentage of students engaging in good hand hygiene could be due to a waning perception of COVID-19 risk. Other factors that could undermine good hand hygiene are the lack of adequate facilities, such as soap and water, in schools. 

Male students were more likely to engage in good hand hygiene in the studied school. This finding is different from other published studies, where females typically reported higher good hand hygiene compared to males [31,38,39]. It has been postulated that females are more likely to follow hand hygiene than males, due to their nature of being less willing to engage in risky activities [40]. However, self-reporting of hand hygiene may be inflated (reporting bias), when compared to observed data, meaning that good or desirable behaviour is more frequently reported than observed [41]. Alternatively, social and cultural norms could play a role in undertaking or engaging in different health activities [42]. Therefore, studies to understand the local context are imperative in developing effective health education strategies. 

In this study, 22.8% of students were identified as good face mask users; this finding was much lower than that reported in other studies [31]. Lower mask use in the study population could be due to “mask tiredness,” coinciding with the relaxing of lockdowns, which might have given a false sense of reduced risk of COVID-19 infection. However, face mask use should be encouraged, due to the recent increase in COVID-19 cases in the region and the Bhutan government has reinforced mandatory use of it. Proper use of face masks has been shown to reduce the spread of COVID-19 [43,44,45,46]. The risk of spread is greater the closer a person is to a source of COVID-19 [43,47]. A study has shown a 62% reduction in the risk of predicted risk of COVID-19 among those who self-reported always using face masks [45]. A rapid systematic review of the efficacy of face masks showed that wearing them could be beneficial in the context of COVID-19 outbreaks [48].

Boarding students were more likely to use face masks consistently. Possible reasons for this include reinforcement by teachers and counsellors in the students residing on the school campus. Studies have shown that repetition and reinforcement play a key role in the sustainability of health education [49]. Therefore, reinforcing face mask use in schools where a large number of students congregate can be beneficial, especially in the event of COVID-19 cases in schools.

We observed a weak, but positive correlation between hand hygiene and face mask use. Other studies have reported a similar correlation between positive knowledge and practice [50,51], and attitude and practice [51]. This means that students engaging in one activity are likely to embrace other positive health activities. Therefore, reinforcing good hand hygiene can increase face mask use and vice versa. Such positive practices are important for breaking the transmission cycle of COVID-19 in the community. Although higher knowledge has been shown to reinforce healthier behaviours [52], according to Blooms, knowledge alone may not be enough to bring changes in habits [53]. Attitudes and learning from role-models are more predictive. Therefore, teachers and parents can play important roles. In this study, boarding students had better face mask use than day students, likely through teachers reinforcing this habit in the school.

In conclusion, the number of students engaging in good hand hygiene and face mask use in this study was quite low. These non-pharmacological preventive measures are important in preventing the spread of the COVID-19 pandemic, especially with the recent increase in COVID-19 cases in neighbouring countries. In addition, long-COVID and reminiscent disease (e.g., thromboembolic events) and chronic infections by SARS-CoV-2 play a significant role and new lineages of SARS-CoV-2 show up with new properties may show spillover effects or mutation in immunocompromised persons with the potential for mutations and newer immuno-evading viruses. Therefore, good hand hygiene and face mask use still have added roles in the COVID-19 pandemic as well as in preventing other diseases [35,54,55]. The hand hygiene results in female students require further research, as most studies have reported good hand hygiene in females, contrary to this study. To motivate good hand hygiene behaviour, health promotion messaging could focus on addressing risk perceptions of COVID-19, which might have shared benefits to promote engagement in additional COVID-19 prevention measures. Finally, increasing the visibility and accessibility of handwashing and hand sanitizing signage and materials in public settings may encourage and facilitate hand hygiene to prevent the spread of COVID-19.

## 5. Limitations of the Study

The results of this study were subject to at least five limitations. First, this was a cross-sectional study and, so, causal inferences could not be established. Second, due to the similar socio-demographic characteristics of the students, most of these factors were not associated with the outcome of interest. This could be possibly addressed by sampling schools from different parts of Bhutan, including schools from urban, semi-urban, and urban areas. Third, the responses were self-reported and, so, may be subject to recall and response bias. This bias could be addressed by objectively measuring hand hygiene and face mask use through compliance studies. Fourth, social desirability may have led to over-reporting of hand hygiene and face mask use by the students than in actual practice. Finally, the respondents were not asked about the frequency and access to soap and water or hand sanitiser, which may influence their hygiene behaviours.

## Figures and Tables

**Figure 1 ijerph-20-01058-f001:**
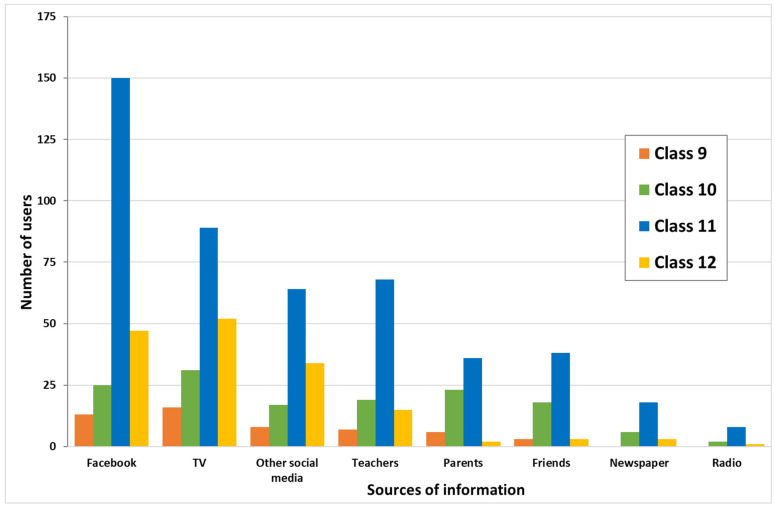
Different sources of information on hand hygiene and mask use reported by the students of Mongar Higher Secondary School, 2022. TV, television.

**Figure 2 ijerph-20-01058-f002:**
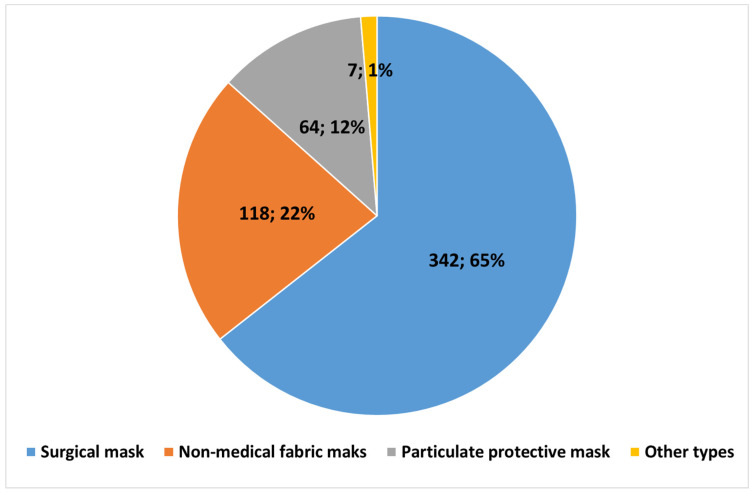
Different types of face masks used by the students of Mongar Higher Secondary School, Bhutan. Note: Actual numbers and percentages.

**Table 1 ijerph-20-01058-t001:** Sociodemographic characteristics of the study population, Mongar Higher Secondary School, Bhutan.

Characteristics		Total		Good Hand Hygiene		Good Face Mask Use	
		N	%	N	%	N	%
Sex							
	Female	282	52.9	79	44.1	55	46.6
	Male	251	47.1	100	55.9	63	53.4
Grades							
	9	53	10.0	24	13.48	12	10.17
	10	45	8.5	11	6.18	8	6.78
	11	283	53.3	90	50.56	71	60.17
	12	150	28.3	53	29.78	27	22.88
Boarder ^£^							
	No	203	38.4	62	34.83	36	31.0
	Yes	326	61.6	116	65.17	80	69.0
Father education							
	NFE *	299	56.3	104	58.1	66	55.93
	Primary	90	17.0	32	17.88	23	19.49
	High school	86	16.2	32	17.88	20	16.95
	Diploma	29	5.5	7	3.9	4	3.39
	Bachelor and above	27	5.1	4	2.2	5	4.2
Mother education							
	NFE *	405	76.1	136	76.4	86	73.5
	Primary	62	11.7	21	11.8	17	14.53
	High school	53	10.0	17	9.6	12	10.26
	Diploma & Bachelor	12	2.3	4	2.2	2	1.71
Father occupation							
	Farmer	292	55.9	103	58.52	64	54.7
	Civil Servant	143	27.4	41	23.3	29	24.79
	PE ^†^	29	5.6	12	6.82	8	6.84
	Driver	13	2.5	4	2.3	5	4.27
	Others	45	8.6	6	9.1	11	9.4
Mother occupation							
	Farmer	275	52.0	94	52.81	62	52.5
	Housewife	166	31.4	59	33.15	38	32.2
	Civil Servant	48	9.1	14	7.87	9	7.6
	PE ^†^	21	4.0	6	3.4	5	4.2
	Others	19	3.6	5	2.8	4	3.4

^£^ Boarder students stay in the school hostel, while day students reside in private accommodation outside school; * NFE, no formal education; ^†^ Private employee.

**Table 2 ijerph-20-01058-t002:** Multivariable logistic regression of good hand hygiene among the students of Mongar Higher Secondary School, Bhutan.

Characteristics		Good Hand Hygiene					
		OR	95% CI	*p* Value	AOR	95% CI	*p* Value
Sex							
	Female	Ref			Ref		
	Male	1.76	1.184–2.445	0.004	1.79	1.230–2.613	0.002
Grades							
	9	Ref			Ref		
	10	0.39	0.163–0.932	0.034	0.4	0.158–0.998	0.05
	11	0.56	0.311–1.023	0.059	0.6	0.322–1.105	0.101
	12	0.66	0.35–1.247	0.201	0.71	0.373–1.381	0.32
Boarder ^£^							
	No	Ref					
	Yes	1.26	0.863–1.828	0.233			
Father education							
	NFE *	Ref			Ref		
	Primary	1.04	0.632–1.694	0.893	1.21	0.594–2.086	0.676
	High school	1.11	0.675–1.828	0.678	1.31	0.428–2.058	0.703
	Diploma	0.6	0.247–1.442	0.252	0.67	0.136–2.107	0.449
	Bachelor and above	0.24	0.11–0.968	0.044	0.35	0.216–3.092	0.9
Mother education							
	NFE *	Ref					
	Primary	1.01	0.576–1.782	0.964			
	High school	1.30	0.506–1.723	0.827			
	Diploma & Bachelor	2.43	0.293–3.343	0.986			
Father occupation							
	Farmer	Ref			Ref		
	Civil Servant	0.74	0.478–1.139	0.170	0.82		0.527
	PE ^†^	1. 3	0.596–2.817	0.514	1.53		0.332
	Driver	0.82	0.245–2.713	0.74	0.76		0.656
	Others	1.01	0.525–1.951	0.971	0.97		0.921
Mother occupation							
	Farmer	Ref					
	Housewife	1.06	0.709–1.59	0.771			
	Civil Servant	0.79	0.406–1.55	0.497			
	PE ^†^	0.77	0.289–2.05	0.601			
	Others	0.69	0.24–1.967	0.485			

^£^ Boarder students stay in the school hostel, while day students reside in private accommodation outside school; * NFE, no formal education; ^†^ Private employee; OR, odds ratio; AOR, adjusted odds ratio; CI, confidence interval; Ref, reference group.

**Table 3 ijerph-20-01058-t003:** Multivariable logistic regression of good face mask use using among the students of Mongar Higher Secondary School, Bhutan.

Characteristics		Good Face Mask Use					
		OR	95% CI	*p* Value	AOR	95% CI	*p* Value
Sex							
	Female	Ref			Ref		
	Male	1.38	0.918–2.084	0.121	1.42	0.934–2.166	0.1
Grades							
	9	Ref					
	10	0.74	0.272–2.006	0.552			
	11	1.14	0.57–2.3	0.705			
	12	0.75	0.349–1.614	0.462			
Boarder ^£^							
	No	Ref			Ref		
	Yes	1.51	0.972–2.342	0.067	1.68	1.001–2.813	0.046
Father education							
	NFE *	Ref					
	Primary	1.21	0.701–2.094	0.491			
	High school	1.07	0.605–1.892	0.817			
	Diploma	0.57	0.19–1.681	0.305			
	Bachelor and above	0.8	0.293–2.2	0.669			
Mother education							
	NFE *	Ref					
	Primary	1.4	0.764–2.57	0.276			
	High school	1.09	0.547–2.156	0.814			
	Diploma & Bachelor	0.74	0.16–3.449	0.703			
Father occupation							
	Farmer	Ref			Ref		
	Civil Servant	0.91	0.554–1.484	0.696	1.19	0.684–2.077	0.535
	PE ^†^	1.36	0.574–3.208	0.487	1.93	0.764–4.853	0.165
	Driver	2.23	0.704–7.041	0.173	2.26	0.706–7.228	0.17
	Other	1.15	0.553–2.402	0.705	1.45	0.678–3.1	0.338
Mother occupation							
	Farmer	Ref					
	Housewife	1.02	0.644–1.615	0.933			
	Civil Servant	0.79	0.364–1.726	0.559			
	PE ^†^	1.07	0.378–3.047	0.894			
	Other	0.92	0.293–2.861	0.88			

^£^ Boarder students stay in the school hostel, while day students reside in private accommodation outside school; * NFE, no formal education; ^†^ Private employee. OR, odds ratio; AOR, adjusted odds ratio; CI, confidence interval; Ref, reference group.

**Table 4 ijerph-20-01058-t004:** Correlation between good hand hygiene and mask use in Mongar Higher Secondary School, Bhutan.

Domain	N (%)	
Hand hygiene		Hand hygiene
Poor	354 (66.4)	
Good	179 (33.6)	
Face mask use		
Poor	415 (77.9)	r = 0.3671
Good	118 (22.14)	*p* value < 0.001

Good: >80% of the total score for that domain.

## Data Availability

Data supporting the results can be obtained upon request from the corresponding author.

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
