# Peer review of "Hand Hygiene, Face Mask Use, and Associated Factors during the COVID-19 Pandemic among the Students of Mongar Higher Secondary School, Bhutan: A Cross-Sectional Study"

_ijerph, 2023, doi:10.3390/ijerph20021058_

Round 1

Reviewer 1 Report

The authors present a study into the hygiene behaviours of pupils at a school in Bhutan in response to the covid-19 pandemic. The findings of the study are very interesting and, I imagine, are very applicable to a much wider setting globally than the single school studied.

Line 11 - remove 'background'

Line 11 reads as though there are 2 halves of separate sentences stuck together and are difficult to read. Could you have a look at re-wording this sentence?

Line 58 - Incorrect use of 'Since'. I would suggest that the article is sent to a native English speaker for language and proof reading. There are many language errors throughout the manuscript that hinders the readers ability to understand the text.

In the methods section, could the study design, study setting and population, study size, and sample size be included in a single paragraph? The information provided covers the details it needs to, however, it looks to be in note form and would be better as a single paragraph.

Could the bars on figure 1 be next to each other rather than overlapping?

The figure legend for figure 1 needs to be re-written. A figure legend should provide enough information and detail for the figure to be understood if the rest of the manuscript was not there. What do the colours for 9, 10, 11, and 12 mean?

Line 182 to 184 - will need to explain what boarder and day scholar students are for countries that do not have this in their education system.

I'd be interested to know why male students had better hand hygiene that female students. Typically, in other parts of the world, women would have better hand hygiene then men.

You mention that information for good hand hygiene was available via various types of media. Have you been able to draw a correlation between the degree of hand hygiene practiced and the source of hand hygiene information? In the UK, there was large differences between hygiene information depending on which media source was reporting it. In most cases the media increased the levels of panic and led to more risky behaviours, ignoring hygiene rules, and mass panic buying of sanitisers.

It would be interesting to repeat this study across several institutions around the world and compare different cultural attitudes towards pandemic hygiene. I suspect that you would see many similarities, but some surprising differences.

It might be better to put the paragraph for limitations above the conclusions within the discussion. You are discussing the limitation and future work of this study - an important part of the discussion. Your conclusion should be finial paragraph in the manuscript that the readers finish on.

Author Response

Reviewer 1

  1. The authors present a study into the hygiene behaviours of pupils at a school in Bhutan in response to the covid-19 pandemic. The findings of the study are very interesting and, I imagine, are very applicable to a much wider setting globally than the single school studied.

Response: Thanks for your time in reviewing our manuscript and providing useful comments that will strengthen it.

  1. Line 11 - remove 'background'.

Response: We have removed ‘background’ from the abstract in line number 11.

  1. Line 11 reads as though there are 2 halves of separate sentences stuck together and are difficult to read. Could you have a look at re-wording this sentence?

Response: We have rephrased the sentence to make it easier to read.

Line numbers 11-12:

Non-pharmacological measures, such as hand hygiene and face mask use, continue to play an important role in the fight against the COVID-19 pandemic.

  1. Line 58 - Incorrect use of 'Since'. I would suggest that the article is sent to a native English speaker for language and proof reading. There are many language errors throughout the manuscript that hinders the readers ability to understand the text.

Response: Thanks for the suggestion, we had it edited by the IJERPH editor and attached the certificate.

  1. In the methods section, could the study design, study setting and population, study size, and sample size be included in a single paragraph? The information provided covers the details it needs to, however, it looks to be in note form and would be better as a single paragraph.

Response: As per the comments of the reviewer, we have restructured the study design, setting, and population under one heading and kept the sample size as it is.

Line numbers 71-74:

Study Design and Setting

We conducted a cross-sectional study among the students of Mongar Higher Secondary School in classes 9 to 12, enrolled for the 2022 academic year. Mongar Higher Secondary School was selected through convenience sampling [30]. The survey was carried out between June and July 2022.

  1. Could the bars on figure 1 be next to each other rather than overlapping?

Response: Thanks for suggesting the change to Figure 1. We have made the suggested changes in the revised manuscript.

  1. The figure legend for figure 1 needs to be re-written. A figure legend should provide enough information and detail for the figure to be understood if the rest of the manuscript was not there. What do the colours for 9, 10, 11, and 12 mean?

Response: We have made the suggested changes as suggested by the reviewer. The new legend of the Figure 1 reads as follows:

Line numbers 156-157:

Figure 1. Different sources of information on hand hygiene and mask use reported by the students of Mongar Higher Secondary School, 2022. TV- television

  1. Line 182 to 184 - will need to explain what boarder and day scholar students are for countries that do not have this in their education system.

Response: We have defined the boarder and day students throughout the revised manuscript to make it clear to international readers: line numbers 136, 141, 170, 172, and 174.

  1. I'd be interested to know why male students had better hand hygiene than female students. Typically, in other parts of the world, women would have better hand hygiene than men.

Response: One reason for such a finding would be related to reporting bias. We have added the following in the revised manuscript.

Line numbers 211-213:

However, self-reporting of hand hygiene may be inflated (reporting bias), when compared to observed data, meaning that good or desirable behaviour is more frequently reported than observed [41].

  1. You mention that information for good hand hygiene was available via various types of media. Have you been able to draw a correlation between the degree of hand hygiene practiced and the source of hand hygiene information? In the UK, there was large differences between hygiene information depending on which media source was reporting it. In most cases the media increased the levels of panic and led to more risky behaviours, ignoring hygiene rules, and mass panic buying of sanitisers.

Response: We did run a correlation between good hand hygiene and different sources of information. However, most of them were not significant.

  1. It would be interesting to repeat this study across several institutions around the world and compare different cultural attitudes towards pandemic hygiene. I suspect that you would see many similarities, but some surprising differences.

Response: Thanks for highlighting this important limitation. Authors are planning to repeat similar studies in other schools across Bhutan to get results that are generalizable. In addition, we have suggested to undertake similar studies in the conclusion of this manuscript.

  1. It might be better to put the paragraph for limitations above the conclusions within the discussion. You are discussing the limitation and future work of this study - an important part of the discussion. Your conclusion should be final paragraph in the manuscript that the readers finish on.

Response: We are happy to do as suggested by the reviewer if the journal permits. We drafted the manuscript based on the journal template.

Reviewer 2 Report

Dear Editor and Authors

many thanks for contributing a reviewer on the manuscript "hand hygiene and facemask use and its associated factors during the COVID-19 pandemic among the students of Mongar Higher Secondary School, Bhutan: a cross sectional study" by Wangchuk, Wangid, Tshering and Wangdi submitted in the IJERPH. 

this is a crafty and slender epidemiological study among high school students that is of high relevance for spreading von SARS-CoV2 from schools to households and the society. The study is purely epidemiological, but gives necessary information about the situation in high schools. However in the discussion section the intentions for future studies could be lined out a little bit deeper. 

I only have some suggestions on the manuscript: 

L.42

Delta does not play a significant role any more. Moreover, Omicron sublineages with higher infectious properties show the demand for non-pharmacological interventions and vaccination. 

L45

Please give some emphasize on the role of hand hygiene that is of much lower significance than aerogenic transmission and therefore the use of masks

L74

Please discuss potential bias concerning the study period. In July 2022 epidemiology was another compared to 2020. This could be described as mask compliance may be lowered due to "mask tiredness" in populations and lower infection rates in these months -after the Bhutan main wave in April 2022 (according to WHO-data).  

Did you provide the questionaire in Dzongkha or English? Is there any potential bias if provided in English? Did yu use other languages?

L185

Again discuss COVID-19 efficacy/efficiancy in hand hygiene in comparison to mask compliance as SARS COV2 is transmitted mainly by aerosols. However, hand washing reduces other infections (e.g. Noro or other gastrointestinal viruses) that could have stressed health systems in addition. 

 L185

Please discuss the gender bias in more detail. Risk assessment in males may have played a role, but in these low numbers this may only be a potential cause for your findings. If there are some "central persons" among girls keeping up "negative" role models findings could be explained by this as well. However, social data was not obtained, but this gives interesting impulses for future research among students. Another explanation is survey fraud due to social desirability or bias by overconfidence (please compare literature on overconfidence in hand hygiene: Lamping and Lengerke et al 2022).  However this bias (subjective ratings in the survey compared to real life behaviour) should be discussed deeper.

L227

Please reconsider that the pandemic is not over as long-COVID and remnescant disease (e.g. thrombembolic events) and chronic infections by SARS-COV-2 play a significant role and new lineages of SARS Cov2 show up with new properties, may show spillover effects or mutation in immunocompromised persons with the potential for mutations and newer immuno-evading viruses.

L219

Please reconsider that you only assessed subjective experiences - not objective ones - as you lined out in the limitation section. Therefore the conclusion should be weaker and prior to a conclusion other studies should be conducted for objectivation, e.g. compliance testing.

L230

Tailoring Hand hygiene for females is susceptible for prejudice on that group. As other data and studies show contrary findings, I suggest to consider alleviating this "hard" conclusion. To my eyes this conclusion is not robust from this single observation without a look for the social components and "leaders"/role-models of that group.

L 223

Knowledge (according to Blooms taxonmy) alone may be not the only key factor. If we ask for smoking risks, most people know that is carcinogenic - but they do it anyway. Thus, attitudes and learning by role-models are even more predictive and teachers and parental compliances may have played a role too.  

Author Response

Reviewer 2

  1. Dear Editor and Authors

Many thanks for contributing a reviewer on the manuscript "hand hygiene and facemask use and its associated factors during the COVID-19 pandemic among the students of Mongar Higher Secondary School, Bhutan: a cross sectional study" by Wangchuk, Wangid, Tshering and Wangdi submitted in the IJERPH. 

This is a crafty and slender epidemiological study among high school students that is of high relevance for spreading von SARS-CoV2 from schools to households and the society. The study is purely epidemiological, but gives necessary information about the situation in high schools. However, in the discussion section the intentions for future studies could be lined out a little bit deeper. 

Response: Thank you for reviewing our paper and providing us with insightful comments. Your comments will help strengthen our paper.

I only have some suggestions on the manuscript: 

  1. 42: Delta does not play a significant role any more. Moreover, Omicron sublineages with higher infectious properties show the demand for non-pharmacological interventions and vaccination. 

Response: We have rephrased to include the Omicron sublineages and removed Delta in the revised manuscript.

Line numbers 42-44:

Despite the rolling out of vaccines, newer strains (e.g., Omicron sub-lineages) are emerging, resulting in new waves of COVID-19 infections [3-11].

  1. L45: Please give some emphasize on the role of hand hygiene that is of much lower significance than aerogenic transmission and therefore the use of masks.

Response: The benefits of hand hygiene and face mask has been added in the revised manuscript as suggested.

Line numbers 46-47:

Face mask use has greater role in reducing COVID-19 due to aerosol transmission of it. But hand hygiene can have added health benefits of reducing other infections. 

  1. L74: Please discuss potential bias concerning the study period. In July 2022 epidemiology was another compared to 2020. This could be described as mask compliance may be lowered due to "mask tiredness" in populations and lower infection rates in these months -after the Bhutan main wave in April 2022 (according to WHO-data).  

Response: The potential bias of study period has been added in the revised manuscript.

Line number 218-222:

Lower mask use in the study population could be due to “mask tiredness,” coinciding with the relaxing of lockdowns, which might have given a false sense of reduced risk of COVID-19 infection. However, face mask use should be encouraged, due to the re-cent increase in COVID-19 cases in the region and the Bhutan government has rein-forced mandatory use of it.

  1. Did you provide the questionaire in Dzongkha or English? Is there any potential bias if provided in English? Did you use other languages?

Response: The questionnaire was in English and authors feel students can understand and answer survey questions in English better than Dzongkha.

  1. L185: Again discuss COVID-19 efficacy/efficiency in hand hygiene in comparison to mask compliance as SARS COV2 is transmitted mainly by aerosols. However, hand washing reduces other infections (e.g. Noro or other gastrointestinal viruses) that could have stressed health systems in addition. 

Response: We have made the following changes in the revised manuscript.

Line numbers 46-47:

Face mask use has greater role in reducing COVID-19 due to aerosol transmission of it. But hand hygiene can have added health benefits of reducing other infections. 

Line numbers 252-254:

Therefore, good hand hygiene and face masks use still have added roles in the COVID-19 pandemic as well as in preventing other diseases [35, 53, 54].

  1. L185: Please discuss the gender bias in more detail. Risk assessment in males may have played a role, but in these low numbers this may only be a potential cause for your findings. If there are some "central persons" among girls keeping up "negative" role models findings could be explained by this as well. However, social data was not obtained, but this gives interesting impulses for future research among students. Another explanation is survey fraud due to social desirability or bias by overconfidence (please compare literature on overconfidence in hand hygiene: Lamping and Lengerke et al 2022).  However, this bias (subjective ratings in the survey compared to real life behaviour) should be discussed deeper.

Response: Gender bias has been discussed in detail in the revised manuscript.

Line numbers 211-213:

However, self-reporting of hand hygiene is inflated when compared to observed data, meaning that good or desirable behaviour is more frequently reported than it is observed [41].

  1. L227: Please reconsider that the pandemic is not over as long-COVID and remnescant disease (e.g. thrombembolic events) and chronic infections by SARS-COV-2 play a significant role and new lineages of SARS Cov2 show up with new properties, may show spillover effects or mutation in immunocompromised persons with the potential for mutations and newer immuno-evading viruses.

Response: We agree with the reviewer that the COVID-19 pandemic is not over yet with recent increase of cases. So we have added the suggested changes in the manuscript.

Line numbers 246-254:

These non-pharmacological preventive measures are important in preventing the spread of the COVID-19 pandemic, especially with the recent increase in COVID-19 cases in neighbouring countries. In addition, long-COVID and reminiscent disease (e.g. thromboembolic events) and chronic infections by SARS-COV-2 play a significant role and new lineages of SARS Cov2 show up with new properties, may show spillover effects or mutation in immunocompromised persons with the potential for mutations and newer immuno-evading viruses. Therefore, good hand hygiene and face masks use still have added roles in the COVID-19 pandemic as well as in preventing other diseases [35, 53, 54].

  1. L219: Please reconsider that you only assessed subjective experiences - not objective ones - as you lined out in the limitation section. Therefore, the conclusion should be weaker and prior to a conclusion other studies should be conducted for objectivation, e.g. compliance testing.

Response: This is an important limitation of this study. We have highlighted the ways to overcome the bias through compliance testing studies.

Line number 268-269:

This bias could be addressed by objectively measuring hand hygiene and face mask use through compliance studies.

  1. L230: Tailoring Hand hygiene for females is susceptible for prejudice on that group. As other data and studies show contrary findings, I suggest to consider alleviating this "hard" conclusion. To my eyes this conclusion is not robust from this single observation without a look for the social components and "leaders"/role-models of that group.

Response: We have revised the manuscript to avoid prejudice on female based on a single study as follows:

Line number 254-255:

The hand hygiene results in female students require further research, as most studies have reported good hand hygiene in females, contrary to this study.

  1. L 223: Knowledge (according to Blooms taxonmy) alone may be not the only key factor. If we ask for smoking risks, most people know that is carcinogenic - but they do it anyway. Thus, attitudes and learning by role-models are even more predictive and teachers and parental compliances may have played a role too.  

Response: We have added the following changes as suggested by the reviewer.

Line numbers 239-244:

Although higher knowledge has been shown to reinforce healthier behaviours [52], according to Blooms, knowledge alone may not be enough to bring changes in habits. Attitudes and learning from role-models are more predictive. Therefore, teachers and parents can play important roles. In this study, boarding students had better face mask use than day students, likely through teachers reinforcing this habit in the school.

Round 2

Reviewer 1 Report

This is a very interesting piece of work that has under gone major corrections. Overall, I have no further concerns. I would ask if line 241 requires a reference 'according to Blooms'?

Author Response

This is a very interesting piece of work that has undergone major corrections. Overall, I have no further concerns. I would ask if line 241 requires a reference 'according to Blooms'?

Response: Thanks for your positive comments and taking the time to provide valuable feedback on our manuscript which helped strengthen it. We have added reference to line number 241.

Page line number 241-244:

Although higher knowledge has been shown to reinforce healthier behaviours [52], according to Blooms, knowledge alone may not be enough to bring changes in habits [53].

Reviewer 2 Report

Dear editors and authors

thank you again for the opportunity to review this manuscript. 

I do not have further comments. 

Happy new year!

Author Response

Dear editors and authors

thank you again for the opportunity to review this manuscript. 

I do not have further comments. 

Happy new year!

Response: Thanks for taking the time to provide valuable feedback on our manuscript which helped strengthen it. We also wish you a very happy new year 2013.